# A Reference Standard Process Model for Agriculture to Facilitate Efficient Implementation and Adoption of Precision Agriculture

Rok Rupnik [1], Damjan Vavpotič [1], Jurij Jaklič [2], Aleš Kuhar [3], Miroslav Plavšić [4] and Boštjan Žvanut [5,*]

1   Faculty of Computer and Information Science, University of Ljubljana, Večna Pot 113,
    1000 Ljubljana, Slovenia; rok.rupnik@infolab-fri.si (R.R.); damjan.vavpotic@fri.uni-lj.si (D.V.)
2   School of Economics and Business, University of Ljubljana, Kardeljeva Ploščad 17, 1000 Ljubljana, Slovenia;
    jurij.jaklic@ef.uni-lj.si
3   Biotechnical Faculty, University of Ljubljana, Jamnikarjeva 101, 1000 Ljubljana, Slovenia;
    ales.kuhar@bf.uni-lj.si
4   Faculty of Agriculture, University of Novi Sad, Trg Dositeja Obradovića 8, 21000 Novi Sad, Serbia;
    miroslav.plavsic@stocarstvo.edu.rs
5   Faculty of Health Sciences, University of Primorska, Polje 42, 6310 Izola, Slovenia
*   Correspondence: bostjan.zvanut@fvz.upr.si

**Abstract:** Agriculture is a sector that today demands even greater efficiency; thus, it relies extensively on the use of precision agriculture technologies: IoT systems, mobile applications, and other digitalization technologies. Experience from a large-scale EU-funded project with a consortium made up of several software companies shows that software companies have a different and unequal knowledge/understanding of agricultural processes and the use of precision agriculture in agricultural processes. This finding coupled with what is known about the standard process model for IT governance (COBIT) triggered the idea of a reference standard process model for agriculture (RSPMA), which we present in this paper. We applied the Delphi technique to assess the RSPMA and evaluate its potential implementation in the area of agriculture. A panel of 20 members from Slovenia, Romania, Croatia, and Serbia was established for the study. The majority of RSPMA elements were identified as appropriate for the use in agriculture by the panel. The study results show that RSPMA is suitable for use in this field.

**Keywords:** precision agriculture; software development; IoT systems; standard process model; business process management; Internet of things

## 1. Introduction

In recent years, agriculture has made demands for ever greater efficiency; thus, it has relied extensively on the use of precision agriculture technologies: IoT systems, mobile applications, decision support systems, and *farm ERP* (ERP—Enterprise Resource Planning), i.e., ERP system adapted for farming. Several papers have confirmed that technologies mentioned improve the efficiency of farm management and other agricultural processes [1–4]. The experience gained in an EU-funded project called AgroIT, which was completed in 2016, reveals that software companies possess different and unequal knowledge/understanding of agricultural processes, the use of precision agriculture in agricultural processes, activities within agricultural processes, and process metrics. As part of that project, five software companies were implementing their own software product or IoT system while integration among them was also implemented. The mentioned wide range of knowledge/understanding created a problem not only for the pilot users of all precision agriculture systems, but also for the software companies that attempted to integrate the software products and IoT systems, which were based on various precision agriculture technologies. Although there are many software products and IoT systems on

the market today, many of them cover quite a narrow functional area, making integration an obvious necessity [2].

This finding, coupled with the expertise on COBIT, i.e., the standard process model for the governance and management of IT in companies, introduced later in this paper, triggered the idea for a standard process model for agriculture. Such a model could become a reference for managing farms and could define other agricultural processes; hence, the proposed model is called the *Reference Standard Process Model for Agriculture*—RSPMA. The first version of the RSPMA presented in this paper is defined on the level of concepts, the relations between them, and a list of processes. We believe that the RSPMA would bring benefits for several target groups: farm managers, other workers on farms, agricultural consultants, and product managers in software companies that develop software or IoT systems for precision agriculture. As it will be reflected from the model, it also covers the implementation of new technologies on farms.

In order to test our confidence in the benefits and suitability of implementing the RSPMA in agriculture, we conducted a study based on the *Delphi technique*. During the study, a panel was established that comprised 20 members of the following profiles: academics from the area of agriculture, agricultural consultants, farm managers, and product managers. The study results show the panel was able to reach a consensus on the RSPMA's implementation in agriculture: RSPMA was assessed by the international panel of experts to be a valid reference standard process model with the potential to be implemented in agriculture and become a useful model for the mentioned target groups.

The paper is structured as follows. The Section 2 introduces the RSPMA, where the background and the fundaments are first presented. After that, related works and target groups for the use of RSPMA are presented. Finally, the Section 2 introduces the conceptual model of the RSPMA and a discussion on the relationships between processes. In the Section 3, the methodology for evaluating the RSPMA is introduced. The Section 4 presents the results. The results are discussed in the Section 5 and conclusions appear in the Section 6.

## 2. Reference Standard Process Model for Agriculture—RSPMA

The idea for RSPMA was triggered during an EU-funded project from the area of implementation of precision agriculture. In the next step it was expanded based on knowledge and competences from areas of IT governance, software development and business process management. Background and fundamentals of RSPMA are introduced below.

### 2.1. Background to the RSPMA

AgroIT is an EU-funded project that was carried out between 2014 and 2016, covering the issues mentioned above with respect to the implementation precision technologies in agriculture. Several types of information systems or other systems were implemented. First, the *farm ERP system for agriculture* to facilitate farm management. The *farm ERP* system is a traditional ERP system for small and medium enterprises, which includes additional modules for livestock, fruit growing, winery etc. [1,2,5–8]. Second, a *decision support system* that uses advanced machine learning methods to support decision-making processes [3]. Third, *IoT systems* with various sensors used to facilitate automatic data collection about various sensor measurements [4,9–11]. The project also covered the implementation of a *cloud integration platform*: all applications and IoT systems were integrated through this cloud integration platform to facilitate the online exchange of data through one single point (rather than peer-to-peer) [10,12]. The overview reveals that that project covered a wide range of precision agriculture technologies, as well as the complex integration.

During the analysis phase of the project, it became apparent that the software partners had different and unequal knowledge/understanding of agricultural processes, the use of precision agriculture in agricultural processes, activities within agricultural processes, and process metrics. In the final phase of the project, the idea emerged to define a reference standard process model for agriculture.

*2.2. Fundaments for the RSPMA*

A reference standard process model is a generic abstract process representation based on a small number of unifying concepts [13] that can be used as a baseline while developing and evaluating particular models [14] and is thus a blueprint of generally accepted best practices, i.e., a reusable and efficient business process upon which organizations can design their own process [15].

The RSPMA is designed and built based on various fundaments. First and foremost, the mission and the concepts of COBIT: to create a standardized reference process model for a particular area, in our case, agriculture. We selected COBIT because it is widely used by IT professionals: IT managers, IT experts, and auditors [16,17]. Another element which influenced the selection of COBIT was the fact that the idea and concepts of CO-BIT have been used for quality improvement and standard process model definition in healthcare [18]. Each COBIT concept used is tailored to a structure suitable for agriculture. Further, additional concepts were included based on our confidence in their benefit. COBIT is briefly introduced in a following section of this paper. Second, the model is triggered by the above-mentioned diversity of knowledge/understanding of agricultural processes, the activities within them, and process metrics, as identified in the AgroIT project. Third, it is based on reference process models' positive influence on the efficiency and effectiveness of processes that can be significantly improved when processes are defined by considering reference process models [19]. Fourth, it is based on the positive influence of the presence of reference process models on the efficiency of software development [20,21].

2.2.1. Business Process Management and Business Process Reference Models

For quite some time, farmers and other participants in agricultural value chains have been under pressure to produce and deliver agricultural products of a required quality and quantity [22]. It is thus no longer sufficient to evaluate only the product and not the process by which it is produced [23]. Many other industries have recognized that competitiveness in hyper-competitive and increasingly regulated markets depends heavily on the ability of companies to continuously improve their processes and execute them well [15,24]. Companies that do not govern their processes (this occurs more often in small and medium companies) are exposed to a number of risks which may affect their competitiveness [23]. Farmers and farm managers are thus equally pushed to manage their production processes [25] by considering the use of modern IT, IoT and other technological innovations [26].

Use of a reference model that is high in quality, i.e., complete, general, usable, understandable, accurate, and easily configurable [15], can bring several benefits. The most obvious benefits are the reduction in time and cost required to design the organization's specific processes, as resources required are kept at an acceptable level. Second, risks are lowered because the reference models have already been validated [19,27,28]. Further, their use can lead to improved identification of weaknesses in existing structures and therefore to better and optimized processes, since they usually capture the business insight of more than one industry player [29]. They also provide a common language to link business processes with other members of the supply chain [28]. In general, both the efficiency and effectiveness of processes can be significantly improved [19] if they are defined based on reference standard process models [30,31].

In addition, the reference process model and its content usually form a bridge between the business and IT domains [29]. They can support efforts to achieve interoperability between various systems and the standardization of data [25]. Standardized processes can enable flexible business models without requiring custom interfaces [32]. Besides, they also represent significant opportunities for software vendors and service providers as they can develop applications that interoperate with other elements of the process or supply chain [32].

The adoption of smart management in agriculture (which includes smart monitoring, planning and control of agricultural processes) is hindered by the lack of interoperability

and data exchange between the wide variety of software and hardware systems from different vendors [25]. A set of best practices in reference process models is the foundation for the implementation of integrated systems [25,33,34].

Although the major agricultural machinery vendors have established their own proprietary platforms, these do not include process models and these proprietary platforms make it difficult to ensure interoperability with other components from other vendors [25]. The lack of business process standardization and endless organizational variations can lead to an expensive and chaotic situation, both from a business and information systems perspective, which drives up costs and is, as such, a barrier to competitiveness and development, as seen in some other industries [32].

We believe that the above discussion justifies our research and further steps in development and implementation of RSPMA.

### 2.2.2. The Efficiency of Software Development Supported by a Standard Process Model

Knowing and understanding a customer's software requirements is the crucial component of software development. Such an understanding also means an understanding of the customer's business processes and the corresponding process elements that need to be automated. In recent years, reference process models have been developed in various fields and have significantly contributed not only to the understanding of processes, but also to their standardization, documentation, and communication [35–37].

The existence of a standard process model for a particular area can improve the efficiency of the software development process (of course, only if the software functionalities have been defined based on that model) and the relevance and usefulness of a final software product [38]. Interoperability between software products is another area that benefits from software development based on a standard process model [32].

### 2.3. Related Works

We could not find any related research on a standard process model for agriculture. A literature review showed that while there are some limited process standardization initiatives in agriculture, these largely focus on technical aspects [4,39].

However, there is a good example of the reference standard process model in IT governance, where COBIT is not only defined on a theoretical level, but is widely defined and used by IT experts. IT governance experts mostly consider COBIT as a framework, but also as a standard process model. Both terms, *framework* and *standard process model*, have the same meaning, although the term *framework* is more appropriate for technical sciences. For that reason, we use the term *standard process model* for RSPMA.

COBIT is a comprehensive framework designed to assist organizations in the governance and management of enterprise IT by maintaining a balance between realizing benefits and optimizing risk levels and resource use [40]. It enables IT to be governed and managed in a holistic manner throughout the entire organization. The success of COBIT in various bigger companies [41] has seen it become a de-facto standard for IT governance in companies and organizations in various economic sectors. The success in reaching such a wide audience is based on the work of highly qualified experts coordinated by the Information Systems Audit and Control Association (ISACA)—an international professional association for IT governance.

The evolution of COBIT has been progressing since the first version in 1996 to the current version in 2019. The research presented in this paper started in 2017, when the latest version, COBIT 2019, was not yet available. Therefore, this paper is based on COBIT 5 [40] and also considers some elements of COBIT 4.1 [42].

COBIT defines a set of generic IT processes. For example, COBIT 5 defines 37 processes divided into governance and management domains [43]. The management domain has four sub-domains: align, plan and organize; build, acquire and implement; deliver, service and support; and monitor, evaluate and assess. For each IT process, inputs/outputs, goals and metrics, key activities, responsibilities, and process maturity levels are defined [40].

Following the release of COBIT 2019 in the year 2018, the authors decided to review whether the differences between the two versions were relevant to RSPMA. The main differences can be summarized as follows [44]:

- In COBIT 2019, there are several factors that influence the design for the governance system of the enterprise (e.g., strategy, risk profile, role of IT, IT deployment methods, threat landscape). Nevertheless, they are comparable to the COBIT 5 enablers, but much more simplified.
- Governance and management objectives: what in COBIT 5 was known as enabling processes in COBIT 2019 are governance and management objectives. In simple words, the names of the processes have been reworded to look like objectives.

Most of these changes were tailored to the specifics of the IT domain. However, the last of the above differences requires special attention, i.e., the specificity of COBIT 2019 to use governance and management objectives instead of simple process names as in COBIT 5. After discussion, we concluded that the terminology currently used in agriculture still refers to the process name and the process objective as different but still related elements. Therefore, we decided to adopt the naming of the processes as in COBIT 5. Last but not least, the design of RSPMA allows the adoption of current and future or previous versions of COBIT, if they prove to be appropriate for agriculture.

COBIT as the Basis for the Standard Process Model for Healthcare

The results of the systematic literature review show that COBIT has previously been used as a framework for the governance and management of enterprise IT in various sectors: financial [45], governmental [46], higher education [47], and healthcare [48]. Although it is a well-organized, systematic, and generic framework, its idea and structure have never been applied to other sectors. Although there are some examples of the use of COBIT in agriculture [49,50], they refer exclusively to the governance and management of IT. The only identified exception was in healthcare where the results of a preliminary study suggested the adoption of COBIT framework elements to represent typical healthcare processes [18]. This study reveals that COBIT's mission *to create a standardized reference process model* in healthcare is emerging.

*2.4. Target Groups for the RSPMA*

When designing a standard process model, regardless of the area for which it is intended for, the group designing it must first decide which target groups will use the model and what will be the benefits of its use. For the target groups, it should become a reference standard process model that will be fully accepted by them. The RSPMA is intended for the following target groups:

- *product managers* in software companies developing software products and IoT systems for precision agriculture;
- *managers and owners* of bigger farms: COBIT is primarily intended for larger companies. In fact, we believe that any standard process model should be tailored to bigger institutions (organizations in general). Smaller institutions should then use it to the extent they believe is suitable. This is considered in the design of the RSPMA; and
- *consultants for agriculture* who help farms achieve better results.

In our opinion, the benefits product managers may expect are as follows:

- the RSPMA is intended to be a common denominator, a kind of *Esperanto*, a *knowledge base*, for the development of software products and IoT systems for precision agriculture. Namely, each RSPMA process is described by the following components: process goals, process metrics, KPIs (*key performance indicators*), and process activities. The RSPMA can guide product managers in defining the functionalities of their products [37,38]; and
- integrations between various software products and IoT systems becomes easier when product managers base the functionalities of software products on the RSPMA [25,32].

We believe that managers and owners of larger farms can expect the following benefits:

- *The knowledge and experience* of agricultural experts and academics could be progressively transferred to the RSPMA to introduce best practices for agriculture [51];
- RSPMA could provide *best practice guidelines* for processes and their activities on farms also covering the use of precision agriculture technologies. This would help managers to ensure that the processes are being carried out according to the best practice [36,37];
- *metrics and KPIs defined for processes* could help managers set goals and conduct monitoring, while such an approach would also help reduce risks [4,37];
- *managers* would have better chances of identifying gaps in process execution and monitoring, which in turn will help them avoid or eliminate such gaps while improving both processes and monitoring [36,37,52]. Managers would also be given guidelines on how to use precision agriculture technologies for those purposes;
- managers would be better prepared for any audits. Auditors and creditors will have more confidence if a particular audited farm is *RSPMA compliant* [53]; and
- in addition to managers, other farm employees may also learn about the processes, metrics and KPIs [51].

*Consultants for agriculture* can use the RSPMA as a knowledge base for their work. The RSPMA will have its own content, yet it is also meant to be open to any other sources, standards, guidelines: in general, to any *source of knowledge*. As such, the RSPMA will represent a gateway to other relevant source of knowledge [11,51,54]. If *product managers* and *consultants for agriculture* use the RSPMA, we can expect that, by default, it will be easier for consultants to become familiar with software products whose development is based on the use of the proposed standard process model.

*2.5. Conceptual Model of the RSPMA*

We decided to present the concepts of the RSPMA and the relationships between them through a conceptual model. The traditional rectangle-arrow technique was selected as the technique for presenting the conceptual model. In order to improve the clarity, the RSPMA is presented via *conceptual sub-models*.

In the diagrams below, arrow labels show the name of the relationship to understand the meaning of the relationship and thereby the relationship between two concepts. The direction of an arrow indicates the direction to read and understand the relationship. Below we introduce the conceptual model through diagrams representing conceptual sub-models, where the concepts and relations between them are explained in text. In this text (descriptions and explanations), the names of *concepts* and the *relationships* between them are written in *italics*.

The first sub-model is the *process description conceptual sub-model* shown in Figure 1. *Process* is a core concept of the RSPMA. *Processes* are *grouped into process modules*, where each process module *belongs* to a particular *area of agriculture*. Grouping of processes is only one view to explain the need for using modules and modularity in the RSPMA. Another view arises from the fact that agriculture encompasses several areas: livestock breeding, fruit growing, wine making, etc. Some *process modules are divided into* process sub-modules because some areas of agriculture contain several sub-areas, e.g., livestock breeding: cattle breeding, pig breeding, sheep breeding etc. *Domain* is a concept that represents the mission of the *process module* assigned to a *domain* and the *hierarchical level*: governance level, management level and implementation level. Each *process module* belongs to one of three *domains*.

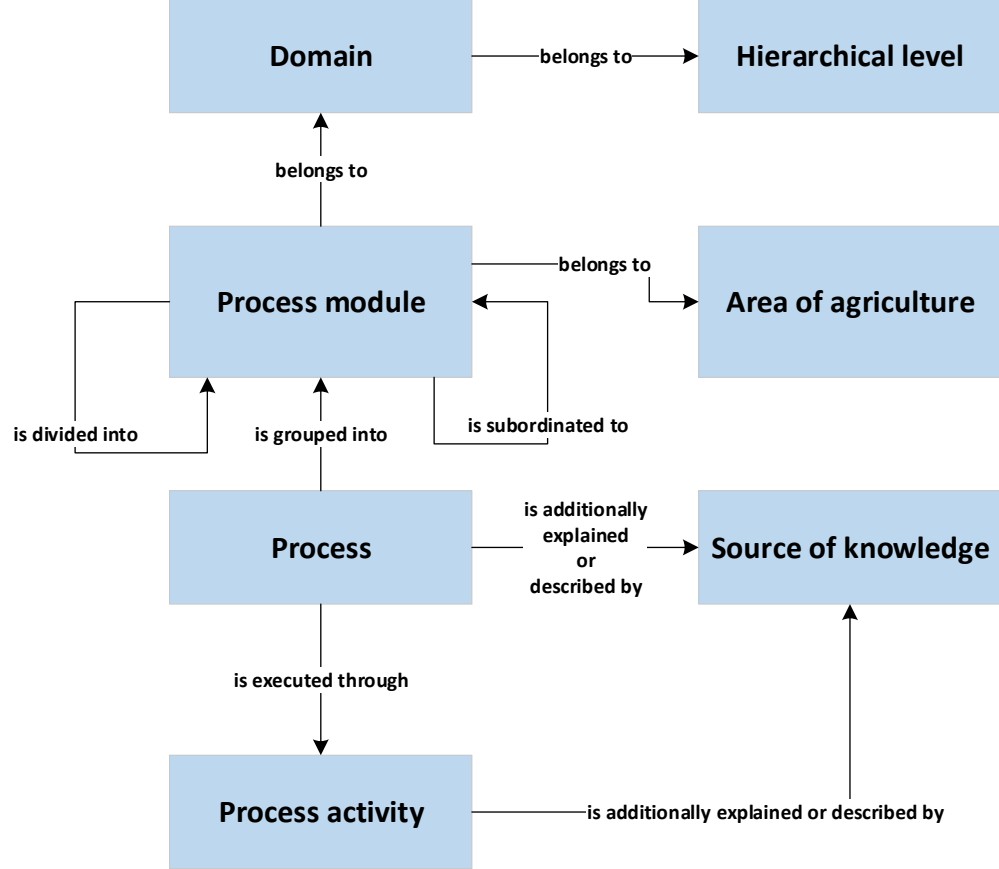

**Figure 1.** Process description conceptual sub-model.

The RSPMA's aim is not to prevail over any existing standard or *source of knowledge* for agriculture: textbook, scientific journal, digital library, standard, etc. The RSPMA is defined and structured to have its own content, but also to be open and, for several of its concepts, to enable a reference to any existing *source of knowledge*. In the conceptual model, this is shown as follows: *process* or *process activity is additionally explained or described by a source of knowledge*.

The second sub-model is the *process risks, contribution, and efficiency conceptual sub-model*, shown in Figure 2. The contribution of a process to the overall outcome of a farm is reflected in *general agricultural economic goals* and *process goals*. Each *process contributes to* one or more *general agricultural economic goals*. *General agricultural economic goal* is a set of economic goals relevant for agriculture and is defined on the RSPMA level. More than one process can contribute to a particular general agricultural economic goal. Each process also contributes to one or more general goals defined by the area of agriculture. General goals defined by the area of agriculture is a set of goals defined on the RSPMA level. Each *process* also *has various additional goals defined* to further describe and explain the *process*. Process efficiency is covered by the following concepts: *key performance indicator*, *process metrics* and *benefit category*. Each *process additionally has various key performance indicators (KPIs) defined* and each *KPI is additionally explained or described by a source of knowledge*. The efficiency of achieving *process goals is measured by process metrics*.

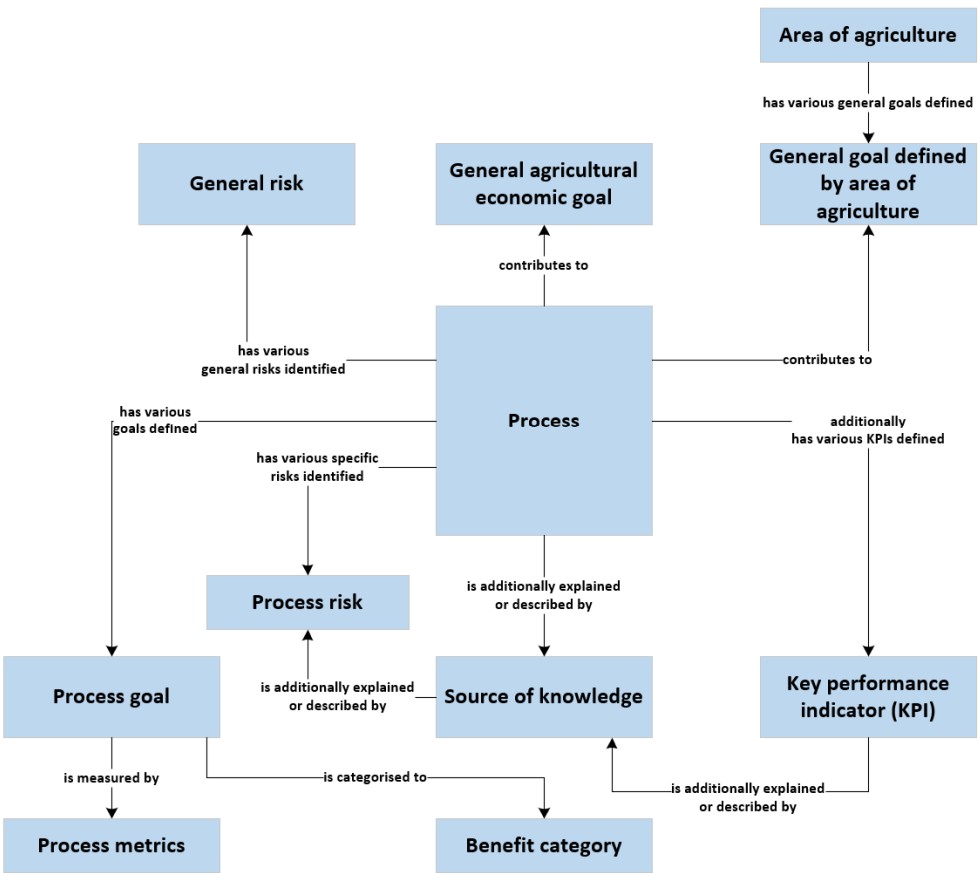

**Figure 2.** Process risks, contribution and efficiency conceptual sub-model.

Risk management is an important issue in agriculture because inefficient risk management can significantly lower the income of a farm [55,56]. Accordingly, the RSPMA also includes concepts to cover risk management and risk assessment. *General risk* is a set of risks relevant for agriculture and is defined on the RSPMA level. Each process *has various general risks identified*. Further, each *process has various specific process risks defined*.

*2.6. RSPMA Processes and the Relations between Them*

The RSPMA has *three domains*, each having a code assigned: *Govern and Monitor* (GM), *Plan and Manage* (PM) and *Implement and Execute* (IE). Each domain has, as already discussed, at least one process module assigned, while in some cases a process module can have a hierarchy of process sub-modules. Figure 3 shows the relationships between domains and the hierarchy of process modules.

Process modules and modularity also have another advantage in the process of creating RSPMA: agriculture is a large compound area that encompasses several areas and such a model can only be built step-by-step where in each step a single process module representing a particular area of agriculture is added. Each domain has a process module called a *common module* (with the code CM). The purpose of this process module is to cover processes that are common to all areas of agriculture and must be performed on any farm, regardless of the areas of agriculture in which the farm operates. The GM domain represents the *governance level* and has only the *common module*. The PM and IE domains also have *common module* and, in addition, process modules for different areas of agriculture. For now, only process modules for the *livestock agricultural area* have been defined for the PM and IE domains. Process module has a unique code which is a concatenation of: domain code, parent's process module codes and process module code (e.g., IE.LS.CB—Cattle Breading).

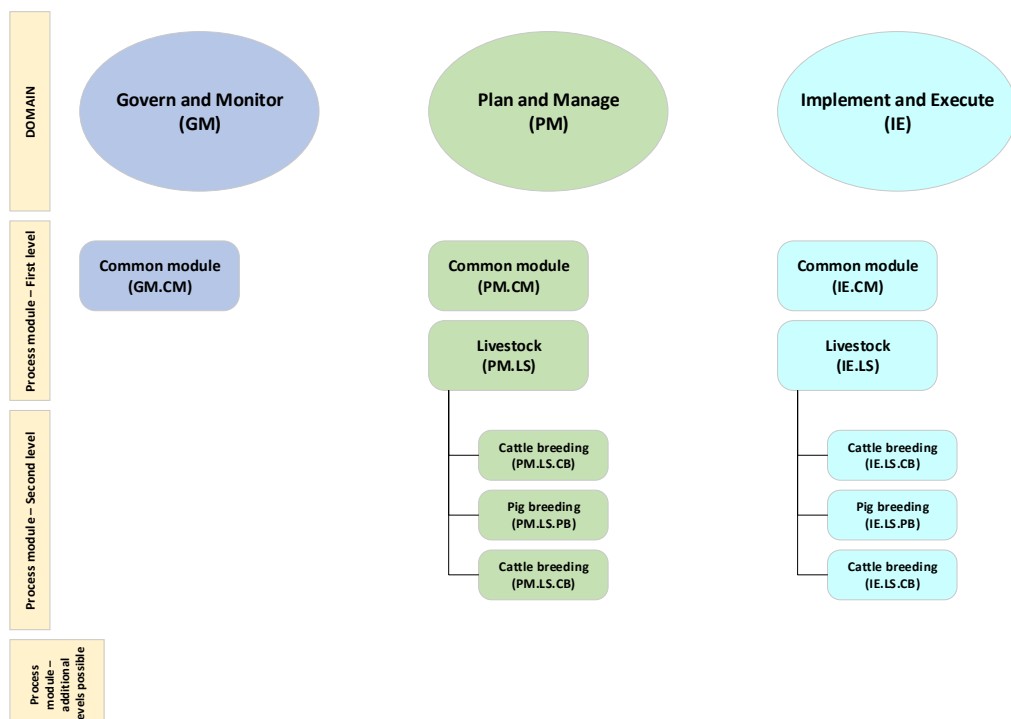

**Figure 3.** RSPMA domains and hierarchy of process modules.

The RSPMA contains an extensive list of processes and we therefore decided to define the naming convention. First, each process has its own unique code. Second, processes are named in an *imperative way* that reflects the process' mission and key goals, for example: *ensure risk governance*, *manage suppliers*, etc.

As mentioned, we followed the top-down division into *governance*, *management*, and *implementation* levels where each level is represented by a 'separate' domain. In such cases, there are always *top-down* and *bottom-up* relations between processes on adjacent levels. *Top-down* and *bottom-up* relations between processes are a *vertical type* of relations. When viewing these relations between processes in a *top-down* direction, a process on a higher level *directs* one or more processes on a lower level. On the other hand, when viewing these relations between processes in a *bottom-up* direction, a process on a lower level *contributes to the implementation of* one or more processes on a higher level.

Figure 4 shows an example of relations between selected processes from all three domains. Information system is the tool needed for every task on farms, so it is not surprising that the relations are quite extensive.

The current list of defined process modules and their processes is presented in Table 1.

As mentioned, in addition to the common modules only process modules for livestock have been defined so far. It is important to note that process sub-modules for various areas of livestock are still to be defined: cattle breeding, pig breeding, sheep breeding, etc.

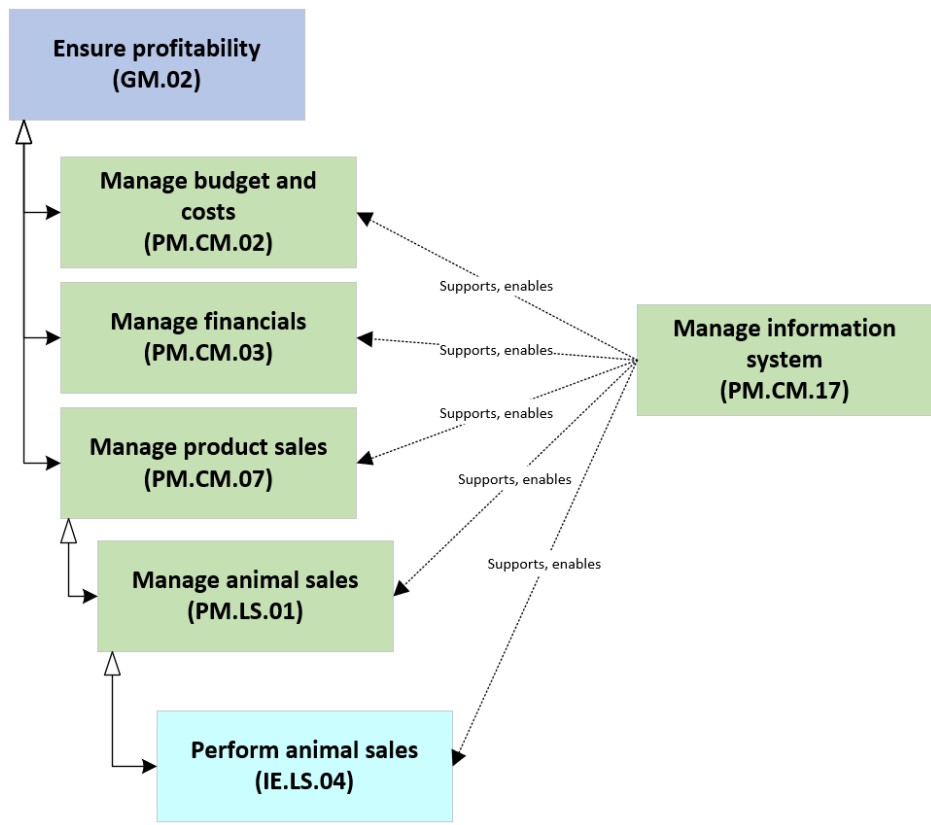

**Figure 4.** Example of relations between processes.

**Table 1.** The list of processes in RSPMA.

| Domain-Process Module | Process |
| --- | --- |
| Govern and monitor (GM)–Common Module (CM) | |
| | GM.01: Define and maintain strategy |
| | GM.02: Ensure profitability |
| | GM.03: Ensure risk governance |
| | GM.04: Ensure machinery and equipment governance |
| | GM.05: Ensure IT and innovation governance |
| | GM.06: Ensure compliance with legislation |
| | GM.07: Enable external and internal control |
| | GM.08: Manage and monitor process definition and change |
| | GM.09: Implement and monitor implementation of strategy |
| Plan and Manage (PM)–Common Module (CM) | |
| | PM.CM.01: Manage implementation of strategy and investments |
| | PM.CM.02: Manage budget and costs |
| | PM.CM.03: Manage financials |
| | PM.CM.04: Manage risks |
| | PM.CM.05: Manage human resources |
| | PM.CM.06: Manage buildings and security |
| | PM.CM.07: Manage products sales |
| | PM.CM.08: Manage suppliers |
| | PM.CM.09: Manage sub-contractors |
| | PM.CM.10: Manage certifications |
| | PM.CM.11: Manage environment and protection |
| | PM.CM.12: Manage energy consumption |

**Table 1.** *Cont.*

| Domain-Process Module | Process |
| --- | --- |
| | PM.CM.13: Manage energy production |
| | PM.CM.14: Manage agricultural machinery |
| | PM.CM.15: Manage equipment |
| | PM.CM.16: Manage IT |
| | PM.CM.17: Manage information system |
| | PM.CM.18: Manage innovations |
| | PM.CM.19: Manage investment projects |
| | PM.CM.20: Manage needs and expectations |
| | PM.CM.21: Manage knowledge and legislation |
| | PM.CM.22: Manage changes based on legislation demands |
| | PM.CM.25: Manage changes based on IT and innovation |
| | PM.CM.26: Manage assets |
| | PM.CM.27: Manage technical capacity |
| | PM.CM.28: Manage internal control |
| Plan and Manage (PM)–Livestock Module (LS) | |
| | PM.LS.01: Manage animal sales |
| | PM.LS.02: Manage animal purchases |
| | PM.LS.03: Manage animals health and veterinary service |
| | PM.LS.04: Manage animal welfare |
| | PM.LS.05: Manage hygiene |
| | PM.LS.06: Manage animal feeding and grazing |
| | PM.LS.07: Manage animal reproduction |
| | PM.LS.08: Manage animal breeding plan |
| Implement and Execute (IE)–Common Module (CM) | |
| | IE.CM.01: Perform internal control |
| | IE.CM.02: Perform farm accounting |
| | IE.CM.03: Perform maintenance of buildings |
| | IE.CM.04: Perform employment and other human resource issues |
| | IE.CM.05: Perform product sales |
| | IE.CM.06: Perform purchases of equipment |
| | IE.CM.07: Perform purchases of agricultural machinery |
| | IE.CM.08: Perform purchases and implementation of software products |
| | IE.CM.09: Perform asset maintenance |
| | IE.CM.10: Perform purchases |
| Implement and Execute (IE)–Livestock Module (LS) | |
| | IE.LS.01: Perform animal feeding |
| | IE.LS.02: Perform animal movements and grazing |
| | IE.LS.03: Preform animal health checking and health treatment |
| | IE.LS.04: Perform animal sales |
| | IE.LS.05: Perform animal purchases |
| | IE.LS.06: Perform animal selection |
| | IE.LS.07: Perform animal reproduction |

## 3. The Methodology for Evaluating Potential RSPMA Implementation in Agriculture

The *Delphi technique* was used to evaluate the RSPMA and assess its potential implementation in the area of agriculture. According to [57], the Delphi technique is widely used in various fields, e.g., business/economics, informatics, healthcare [58] and also in agriculture [59] to elicit and refine group judgements on a particular problem/topic to reach a consensus among experts. The evaluation entailed four phases: *planning*, *setting up the panel*, the *Delphi consensus process*, and *presenting the results* (interpreting the final data) [60]. The Delphi consensus process can include both qualitative and quantitative data [61]. The first round of the Delphi consensus process is usually based on open-ended questions to solicit the opinions of the panel lists, while subsequent rounds are used to

determine and hopefully reach the desired level of consensus [61]. When possible, the level of consensus is assessed quantitatively [58], as in our study.

The group for implementing and monitoring the study was made up of six members, the co-authors of this paper: researchers and experts in the fields of agriculture, business processes, and informatics. At a beginning, a panel of international experts in the area of agriculture was assembled. In the planning phase, we decided to include four expert profiles in the field of agriculture: consultants (CO), farm managers (CEO), product managers (PM), and academics (AC). To be eligible to participate in the panel, according to the expert profile candidates should possess proper experience, competencies, and references. In addition, the academics had to be enrolled as professors in the field of agriculture and have good references in terms of working with large farms, e.g., through projects, counselling, etc. Initially, a potential list of 25 eligible candidates was made, most of whom were contacted in person and by email to participate in the study. Initially, only 12 candidates responded to the invitation and were willing to participate in the study. Due to the relatively low response rate, a snowball sampling technique was used where the experts who had responded to the invitation were asked to propose other possible candidates. After several iterations, a panel of 20 experts was established. The panelists are from the following countries: Slovenia, Romania, Croatia, and Serbia. According to [62], 15 or more experts can maximize the reliability and minimize the group error in the degree of consensus. Table 2 shows the number of experts by profile and by country.

**Table 2.** Number of panel members by profile by country.

| Panel Member Expert Profile | Country | No. of Panellists |
|---|---|---|
| Consultant | | |
| | Slovenia | 2 |
| | Romania | 2 |
| | Serbia | 1 |
| | Croatia | 1 |
| Farm manager | | |
| | Slovenia | 1 |
| | Serbia | 1 |
| | Croatia | 2 |
| Academic | | |
| | Slovenia | 1 |
| | Romania | 4 |
| | Serbia | 2 |
| | Croatia | 1 |
| Product manager | | |
| | Slovenia | 2 |

Both Delphi rounds were conducted between March first and 30 June 2019. In the first round, open-ended live interviews were conducted with five selected panel members: one product manager, two consultants and three academics as core panel members for the live interviews. An open-ended questionnaire (see Appendix A) was used as a guide. The questionnaire was sent to these members prior to the interview with the instruction that the questionnaire would only guide the discussion and should not be completed. The interviews lasted to two hours, depending on the discussion between the researcher and the panel member. Special attention was taken to avoid reflexivity and the interviewer made a special effort to ask neutral questions without giving explicit or implicit hints to the interviewee [63]. After each interview transcriptions were made. Finally, the researchers analyzed the transcripts together. Since the identified core categories were actually elements of a SWOT analysis (strengths, weaknesses, opportunities, and threats), we decided to use a combination of the Delphi technique and SWOT analysis. Similar approach has been successfully used in previous studies [18,64,65]. Therefore, the outcome

of the first round was a SWOT analysis and its elements. Figure 5 shows key elements of our Delphi procedure.

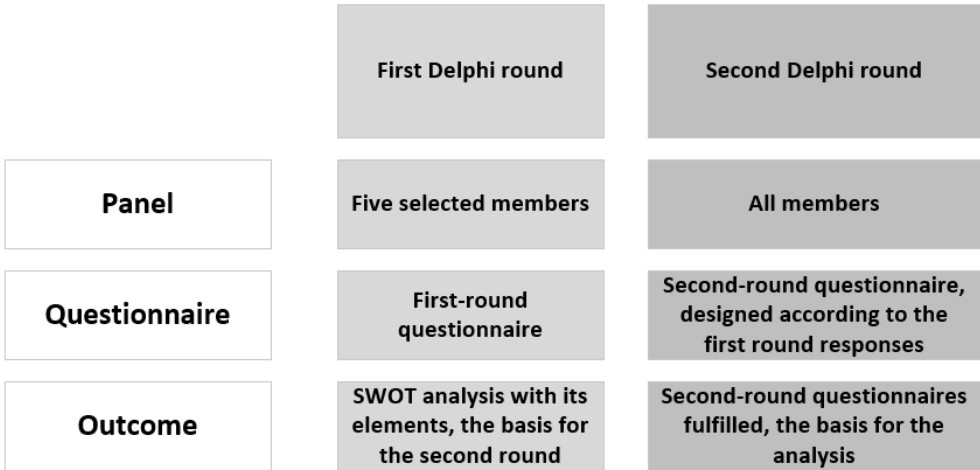

**Figure 5.** The schema for Delphi rounds.

The results of the qualitative analysis were used as a reference while designing the closed-ended questionnaire used in the second round. This questionnaire consisted of guidelines for completing the questionnaire and four sections representing the identified strengths, weaknesses, opportunities, and threats of the SWOT analysis. All panel members were required to indicate the extent to which they agreed with each item (as indicated in Table 3) on a nine-point Likert scale, ranging from 1 (I completely disagree) to 9 (I completely agree). Two criteria for a consensus were adopted from [58]: (1) $\bar{x} \geq 7$ on the nine-point scale (agreement) or $\bar{x} \leq 3$ (disagreement); and (2) at least 75% of responses rated 7–9 (agreement) or 75% of responses rated 1–3 (disagreement). As the consensus was reached after the second round, we did not conduct another round.

**Table 3.** Identified elements by SWOT groups in the first round and the corresponding mean value, SD, minimum, maximum, proportion within the range of the responses after the second round.

| SWOT Group | Identified Element | $\bar{x}$ | SD | Min | Max | Proportion within a Range * |
|---|---|---|---|---|---|---|
| Strengths | Facilitates the design and customization/renovation of processes on farms | 8.3 | 0.9 | 6 | 9 | 95% |
| | Is a sound, clear and understandable presentation of key elements for the design and customization/renovation of processes | 8.3 | 0.9 | 6 | 9 | 95% |
| | Facilitates the implementation of success factors and criteria for effectiveness | 8.3 | 0.9 | 6 | 9 | 95% |

**Table 3.** *Cont.*

| SWOT Group | Identified Element | $\bar{x}$ | SD | Min | Max | Proportion within a Range * |
|---|---|---|---|---|---|---|
| | Concise presentation of key elements for the design and customization/renovation of processes (all information in one place) | 8.2 | 0.9 | 6 | 9 | 95% |
| | Reduces the risks in agriculture if success factors and criteria for effectiveness are taken into account | 8.2 | 0.7 | 7 | 9 | 100% |
| Weaknesses | Risk of yet more unnecessary documentation | 2.3 | 1.2 | 1 | 5 | 85% |
| | Processes are not represented by the process workflow | 2.7 | 1.3 | 1 | 5 | 75% |
| | Lack of comprehensive guidelines for implementing improvements/changes to processes of farms | 2.8 | 1.6 | 1 | 6 | 80% |
| | Such process presentations are too general | 2.8 | 1.4 | 1 | 7 | 80% |
| Opportunities | Systematic and harmonized definition of roles and their competencies in the processes | 8.2 | 0.9 | 7 | 9 | 100% |
| | Possibility of improving information systems and information systems integrations in the field of agriculture | 8.1 | 0.9 | 6 | 9 | 95% |
| | Facilitates the harmonization of the processes with (EU & government) regulation requirements | 7.9 | 1.1 | 4 | 9 | 95% |
| | Possibility of harmonizing information systems of other stakeholders participating in the processes | 7.8 | 1.2 | 5 | 9 | 85% |

**Table 3.** *Cont.*

| SWOT Group | Identified Element | $\bar{x}$ | SD | Min | Max | Proportion within a Range * |
|---|---|---|---|---|---|---|
| | Harmonization of quality assurance systems with farms and other stakeholders participating in the processes | 7.5 | 1.3 | 5 | 9 | 75% |
| | Harmonized teaching and self-learning material prepared by experts in different fields of agriculture | 7.1 | 2.4 | 1 | 9 | 75% |
| | The influence of lobbies on the process presentations and description | 2.2 | 1.2 | 1 | 5 | 85% |
| | Potential risk of inadequately prepared presentations and descriptions of processes | 2.5 | 1.1 | 1 | 6 | 90% |
| | Possibility of outdated process presentations | 2.5 | 1.1 | 1 | 5 | 90% |
| Threats | Large amount and high complexity of processes' presentations/descriptions if all particularities of different fields of agriculture are considered | 2.7 | 1.4 | 1 | 6 | 80% |
| | Local and regional particularities might not be adequately addressed and covered | 2.8 | 1.2 | 1 | 5 | 75% |
| | Competitors will have access to the process model | 2.8 | 1.6 | 1 | 7 | 80% |

Legend: $\bar{x}$—mean value; SD—standard deviation; min—minimum; max—maximum; * Proportion within the range of 7–9 was computed if $\bar{x} > 5$, or within a range of 1–3 if $\bar{x} < 5$.

## 4. Results

### 4.1. First Round

In the continuation, some relevant quotations, identified in the first round, are presented. The five selected panelists reported several strengths of RSPMA, for example:

- *"I like this process model because it can help present complex agricultural processes in a sound, clear and understandable way"*.
- *"On many occasions we had to redesign our processes. If we had access to the processes represented in this way, it would be easier for us to identify the critical elements for their redesign"*.

In addition, several examples of opportunities have been noted:

- "*Such a process representation has significant potential in integrating different information systems*".
- "*Such a process representation with all these elements can be a valuable tool to help us harmonize with national and EU legal requirements*".

Several potentially negative aspects of RSPMA were also mentioned in the interviews. Here are two examples of quotations that represent its weaknesses:

- "*My biggest concern is that by using such a model more and more papers (i.e., unnecessary documentation) would be produced*".
- "*In my humble opinion, such process representations are too generic*".

Below there are two examples of quotes representing threats:

- "*What if the process descriptions were not properly prepared. That is my big concern*".
- "*Someone has to make sure that the processes are presented correctly*".

The detailed analysis revealed five key strengths, four weaknesses, six opportunities, and six threats. These are shown in Table 3—please, refer to columns SWOT group and Identified element.

*4.2. Second Round*

Table 3 presents the results of the data analysis of the second round. Consensus was reached for all elements in the SWOT group *Strengths* as both consensus criteria were met for all elements. Both mean values and the proportion within the range for the elements of the group *Strengths* were considerably higher than the minimum defined in the two previously mentioned criteria. Similar results were obtained in the SWOT group *Opportunities*. The elements "*Harmonization of quality assurance systems with farms and other stakeholders participating in the processes*" and "*Harmonized teaching and self-learning material, prepared by experts in different fields of agriculture*" had the proportion within the range of 75%, which represents a marginal consensus.

The mean values of the responses for all the identified elements of the SWOT group *Weaknesses* were less than 3, indicating disagreement with the identified weaknesses. Panel members were consistent in their responses, except for the element "*Processes are not represented by the process workflow*". This element had a proportion within a range of 75%, indicating a marginal consensus. Similar results were also obtained in the SWOT group *Threats*, where mean values of the responses were below 3, indicating disagreement with the identified threats. A consensus was reached for all identified elements. Slightly inconsistent responses were identified in elements "*Local and regional particularities could not be adequately addressed and covered*" and "*The competitors will have access to the process model*", where the proportion within the range was 75%.

**5. Discussion**

The RSPMA presented and evaluated in this paper is the first version in what we believe will become an evolution of versions where each version makes some progress towards better fulfilment of the RSPMA mission: to become a reference standard for processes in agriculture to improve the work of farm managers, farm engineers, software companies that produce software for agriculture, and others.

As this is the first version, we know that the model has potential for changes and improvements, some of which have already been identified during this study. For example, the list of processes that are defined is quite extensive and some processes could be merged without losing sight of their mission and goals; for example: *Manage IT* and *Manage information system*. Further, the structure of the process description (see Appendix A), which is based on the conceptual model presented (the concepts and relations between them), is also the first version with options for changes and improvements. As mentioned, for now only a process module for livestock is covered by the RSPMA, and when additional modules are added the need for new or updated structures will appear.



We are aware that the RSPMA can only be properly developed through a large-scale international project involving several teams of experts from different areas of agriculture. The development of COBIT, for example, has now been going on for almost 20 years and shows that such standard process models are constantly evolving based on the coordinated work of different expert groups and that changes are based on observations and experience of a standard's use in practice.

The presentation of processes as *process workflows* is always an issue in such standards. COBIT, for example, does not include process workflows. We must be aware that the RSPMA is a reference standard for the governance level, while process workflow definitions are a subject of the management level. Discussion with panel members about this issue indicated that process workflows might only confuse farm managers and that, in many cases, it would be very difficult to reach consensus on workflow. A group working on the framework for the governance of healthcare, for example, came to the same conclusion (based on discussions with various experts and managers) and the framework thus does not include process workflows [18]. We believe that other RSPMA concepts (goals, activities, metrics etc.) are sufficient. However, we do not exclude the possibility of adding process workflows to RSPMA in the future.

It is obviously too early to say that the RSPMA is suitable as a *Harmonized teaching and self-learning material* because, but not only because, the panel members did not reach a consensus to recognize this as an opportunity. We believe that the RSPMA has a chance of becoming one, but only if it is made available through a dedicated website that is well structured and has smart and effective search feature and a nice and effective user interface. A website would also be an appropriate tool to promote openness, availability, and dissemination of RSPMA. By definition, a website allows links to other sources, as has already been discussed for the RSPMA conceptual model.

One of the interviewed farm managers commented as follows: "*Local and regional particularities should be adequately addressed and covered*". This comment undoubtedly requires further attention in the development of the RSPMA. Not all panel members were consistent in declaring that this is not a threat to the use of RSPMA in agriculture. This issue deserves attention in the future.

One member of the panel made an interesting remark: "*Competitors of agricultural software, agricultural equipment and machinery will all have access to the process model*". He added that it is essential for the governance and administration of the RSPMA that it should not be financed by the sponsoring companies, in order for the model to remain independent. In the conversation, it was agreed that a membership fee and EU/government contributions are probably appropriate ways of funding. Nevertheless, the panel member did not believe that this was a *threat* to the model.

*Limitations of the Study*

Even though our findings indicate that the RSPMA is suitable and implementable, three limitations of this study must be acknowledged. First, the RSPMA was not tested in practice, but was evaluated by considering experts' opinions. At this stage, it was not possible to test it in real organizations. In fact, to implement the RSPMA in practice requires considerable financial/human resources, as well as organizations willing to test the proposed standard process model. Moreover, the results of applying the RSPMA are not necessarily immediately visible when implemented in real organizations. To overcome this problem, relevant experts with considerable practical experience from different countries were included in the study to express what they would expect from the hypothetical use of the RSPMA in practice. Another limitation is the number of members of the panel. Although considerable effort was made to increase this number, many experts did not respond to the invitation or were unwilling to participate. Nevertheless, the final number is in line with *Delphi methodological recommendations*. Third, we currently have only a structure for the RSPMA, but the processes are not yet described based on this structure.

## 6. Conclusions

We presented the first version of the RSPMA, which was evaluated by an international panel of experts, as a valid process reference model with the potential for implementation in agriculture. The idea for the RSPMA emerged in the final stages of a large-scale EU-funded project, where several software companies had diverse and unequal knowledge/understanding of agricultural processes, activities within agricultural processes, and process metrics. The diversity and unequal knowledge/understanding posed a problem not only for the pilot users who were using all software products and IoT systems, but also for the software companies that were trying to integrate the software products and IoT systems based on different precision agriculture technologies. The RSPMA is presented on the level of concepts, the relations between them, and a list of processes, and is accordingly not yet ready for the pilot use in practice. The RSPMA and its possible implementation were evaluated using the *Delphi technique* with the help of 20 panel members. The results show that, for the majority of elements, the panel reached consensus on the implementation of the RSPMA in agriculture.

Due to the mentioned study limitations, we would like to further develop the RSPMA through a large-scale international project, considering several elements that are critical to the success of RSPMA. First, the cooperation of reputable research and professional agricultural organizations to establish expert teams for all areas of agriculture. Second, implementing a web-based application to support the work of the expert teams. Third, to establish a website and mobile application to facilitate RSPMA learning and training. Fourth, the execution of a pilot project for pilot implementation of the RSPMA on 20 to 40 large farms around Europe and five to 10 large farms on other continents.

**Author Contributions:** Conceptualization, R.R., A.K. and B.Ž.; methodology, R.R. and B.Ž.; formal analysis, R.R., M.P. and B.Ž.; investigation, J.J., D.V., R.R., M.P. and B.Ž.; resources, R.R. and M.P.; data curation, R.R.; writing—original draft preparation, all authors; writing—review and editing, all authors; visualization, R.R.; supervision, D.V.; project administration, R.R. All authors have read and agreed to the published version of the manuscript.

**Funding:** This research received no external funding.

**Institutional Review Board Statement:** Ethical review and approval were waived for this study. The members of the panel participated voluntary and provided their independent expert opinion about the RSPMA.

**Informed Consent Statement:** Informed consent was obtained from each participant involved in the study.

**Data Availability Statement:** Data is available by the authors (R.R.) upon reasonable request.

**Conflicts of Interest:** The authors declare no conflict of interest.

## Appendix A

### Open-ended questionnaire for the first Delphi round

In the first part of the questionnaire the list of processes was presented. The list is presented in Table 1.

**The structure of process description—Basics**

| Domain Code.counter | | Domain: | |
|---|---|---|---|
| Process Description | | Area of Agriculture: | |
| Process mission | | | |

**The structure of process description—Process activities**

| Activity | Source of Knowledge |
|---|---|
| | |
| | |

**The structure of process description—Process contribution**

| General Agricultural Economic Goals |
| --- |
|  |
|  |
|  |

| General Goals Defined by Area of Agriculture |
| --- |
|  |
|  |
|  |

**The structure of process description—Process effectiveness**

| Process Metrics |
| --- |
|  |
|  |
|  |

| Key Performance Indicators |
| --- |
|  |
|  |
|  |

| Process Goals |
| --- |
|  |
|  |
|  |

| Benefit Category |
| --- |
|  |
|  |
|  |

**Questions/points for the interview**

| Question | Sub-Question |
| --- | --- |
| Does the field of agriculture need such a reference standard process model? | |
|  | Why, in your opinion, does agriculture need such a model? |
|  | Would such a model have a positive influence on farm management and decision-making? |
|  | Which profiles would, in your opinion, use process descriptions? |
|  | Which profiles would, in your opinion, use process descriptions? |
|  | Which profiles would, in your opinion, use process descriptions? |
| What do you think about the structure of the process descriptions? | |
|  | Is it clear and easy to understand? |
|  | What would you change in the structure of the process description and why? |
| Let's say that all processes would be fully described based on the structure introduced. In which situations would you use it and why? | |
| What in your opinion would be the proper media for the use of such a model? (A book, PDF files, web application, mobile application etc.) | |
| Which elements of farm management would benefit from such a model and why? | |

**Questions/points for the interview—special questions for particular profiles**

| Question (Profile) |
| --- |
| Consultant: Which sources have you used so far for your self-education? Which sources have you used as a reference to pass on to farm managers or their key engineers? |
| CEO: which sources and references have you used so far to help you manage your farm? How did you perform self-education? |
| CEO: How do you know now that you are doing the right things and in the right way in managing your farm? |
| CEO: How do you follow novelties and progress in areas like IT, ICT and the IoT? How do you make decisions regarding which software and ICT equipment you need to purchase? |

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
