# Peer review of "A Reference Standard Process Model for Agriculture to Facilitate Efficient Implementation and Adoption of Precision Agriculture"

_agriculture, doi:10.3390/agriculture11121257_

Round 1

Reviewer 1 Report

The authors presented RSPMA (a reference standard process model for agriculture), which was evaluated by a panel comprising of 20 members from 4 countries. The authors used a Delphi technique to evaluate RSPMA. The reviewer has some concerns

  1. Firstly, there are many typos and languages issues in this article.

In the abstract section, line 20-21, the sentence has to be restructured.

In the introduction section, line 41, IoT systems which=> IoT systems, which

In the introduction section, line 54, om => on

Section 2.3.1, line 203, the sentence has to be restructured

There are so many typos, so moderate English changes are required.

  1. In addition to the numerous English typos, the authors need to recompile the file. For example, page 7, line 280, there is an error indicating, reference source not found.

  1. In Section 2.3, the authors mentioned the improved version of COBIT, with COBIT 2019 been the latest. It is vital to demonstrate why the authors chose the COBIT 5 and COBIT 4.1 instead of the improved version. To the best of my knowledge, COBIT 5 was introduced in 2012. The authors also need to demonstrate the key differences in COBIT 2019 and COBIT 5, and how the difference will affect their work.

  1. The authors mentioned that COBIT has only been used in financial, governmental, higher education and healthcare. This is wrong because COBIT has also been used in agriculture. See reference below. How does your work differ from the works in the literature?

Othman, Marini, et al. "COBIT principles to govern flood management." International journal of disaster risk reduction9 (2014): 212-223.

Damanik, Arbaiti, and Asep Fajar Firmansyah. "Strategies to Improve Human Resource Management using COBIT 5 For Data and Information Centre of Ministry of Agriculture of Indonesia of Republic." 2018 6th International Conference on Cyber and IT Service Management (CITSM). IEEE, 2018.

Reviewer 2 Report

The authors in this paper proposed a model called the reference standard process model for agriculture – RSPMA for managing farms.

The research work is interesting and the paper is written very well. Few points that require consideration are given below.

IN abstract on line 20 write the name of countries instead of generally saying ‘from four countries’.

Amend list of  keywords by removing —>  implementation of,  precision agriculture; development of IoT systems; adoption of precision agriculture; standard; business’, and add Internet of Things.

In introduction section on line 31 for ‘several papers’, cite those papers, and if [1,2] are the references then remove several.

For “2. Reference Standard Process Model for Agriculture – RSPMA”, write introductory sentence before sub section 2.1.

Under list of references better to update some very old with the recent literature.

such as 

[26, 36, 39, 57, 59]

Reviewer 3 Report

I recommend that the abstract focuses more on the research findings than on general statements.

The introduction well defines the importance of the article

Figure 1 and 2 I recommend graphic redesign, it reduces the overall quality of the text. At higher resolution, the text is of lower quality.

I have no major comments on the literature review, it is well done, I would just recommend expanding the views from a process management perspective to include, for example, the texts below to make the authors' broad range more apparent:

Salierno, G.; Leonardi, L.; Cabri, G. The Future of Factories: Different Trends. Appl. Sci. 2021, 11, 9980. https://doi.org/10.3390/app11219980.

Rolinek, L.; Kopta, D.; Plevny, M.; Rost, M.; Kubecova, J.; Level of Process Management Implementation in SMEs and Some Related Implications. Transformations in Business and Economics 2015, 14, 360-377.

Gorzelany-Dziadkowiec, M. COVID-19: Business Innovation Challenges. Sustainability 2021, 13, 11439. https://doi.org/10.3390/su132011439

Line 280 Error

I am not convinced of the fundamental importance of most of the schemes. I recommend omitting some of them, they do not contribute substantially to the paper.

The methodology section is very well described, where the authors appropriately cite prominent authors who have used the Delphi method.

The results need to be expanded, they have 1.5 pages out of the full text of 21 pages. The 1st page is occupied by a swot table where the identified elements are expressed in descriptive statistics, without added value. At the same time, the Delphi method is definitely not focused on quantity, so I don't understand the reason to interpret the results using: standard deviation; minimum; maximum; etc. Especially when the outputs are from panels of only 20 members.

I strongly recommend expanding on the concepts and possibly including other parts of the research in the results.

Round 2

Reviewer 3 Report

The authors have appropriately addressed most of the comments and have significantly improved the quality of the text. I have no further suggestions for improvement.